# At-Home Blood Pressure Measurements Provide Better Assessments of Orthostatic Hypotension in Parkinson’s Disease

**DOI:** 10.3390/jpm13091324

**Published:** 2023-08-28

**Authors:** Chathurini V Fernando, Sarah Osborn, Malcolm Horne

**Affiliations:** 1Florey Institute of Neuroscience and Mental Health, Parkville, VIC 3010, Australia; chathurini.fernando@florey.edu.au; 2The Bionics Institute, East Melbourne, VIC 3002, Australia; sosborn@bionicsinstitute.org; 3Department of Medicine, University of Melbourne, St Vincent’s Hospital, Fitzroy, VIC 3065, Australia

**Keywords:** Parkinson’s Disease, orthostatic hypotension, hypertension, cardiovascular dysregulation, autonomic dysfunction

## Abstract

Orthostatic hypotension (OH) is common in Parkinson’s Disease (PD). It is intermittent, exacerbated by stressors including meals, medications, and dehydration, and frequently is unrecognized. Although intermittent, assessment is usually by a single “in clinic” BP measurement. This study examines whether 10 home measurements are more sensitive in detecting OH than a single “in clinic” measurement. Participants (44 people with PD and 16 controls) were instructed to measure lying and standing BP at home. BP was measured on five consecutive days upon waking and before bedtime. Symptoms were also assessed using the Movement Disorder Society United Parkinson’s Disease Rating Scale and the Non-Motor Questionnaire. While a postural drop in systolic BP (≥20 mmHg) was recorded “in clinic” in thirteen of the forty-four PD participants, a postural drop was found in at least one of the ten home measurements in twenty-eight of the forty-four participants. Morning hypertension and variability in lying systolic BP was more common in these subjects than in those without a postural drop or the controls. A greater number of measurements of lying and standing BP are more likely to reveal orthostatic hypotension, variation in systolic BP, and hypertension than a single office measurement in people with PD.

## 1. Introduction

Orthostatic hypotension (OH) is common in Parkinson’s Disease (PD), with a prevalence of between 30% and 50% [1,2,3]. OH is important because it leads to impaired cerebral perfusion [4], resulting in well-known symptoms [5] of light headedness, dizziness [6], loss of consciousness, and falls [7,8,9], and has been linked to impaired cognition [10,11,12,13,14,15,16,17] and mortality [18,19]. 

The differences between the pathophysiology of OH in PD and many other causes of OH can be understood by first reviewing the normal physiological response to an orthostatic challenge. Transferring from lying to standing shifts ~700 mls from the central compartment to lower extremities (~500 mL) and pelvic regions (~200 mL) [20,21,22] decreases central venous pressure, which is sensed by cardiopulmonary baroreceptors, resulting in reduced baroreflex signaling in the brain stem, which decreases vagal nerve activity and increases sympathetic activity and the release of noradrenaline. This, in turn, increases peripheral resistance, heart rate, and contractility [20,21,22]. Thus, there are both central and peripheral mechanisms of regulation: “central mechanisms” refer collectively to the brainstem and cortical structures that regulate autonomic function and include the dorsal motor nuclei of the vagus, the medullary reticular formation, the locus coeruleus [21,23] and insular cortex [24], and “peripheral mechanisms”, referring collectively to vagal and the pre- and postganglionic sympathetic control of end organs. It is important to note for later discussion that central mechanisms selectively control the perfusion of specific vascular beds, depending on their physiological demands. When central pressures are low, this same mechanism prioritises perfusion of the brain, heart, and kidneys over perfusion of other vascular beds. For example, food ingestion is followed by splanchnic vasodilation and the pooling of splanchnic blood, which activates the baroreflex mechanism to maintain normal BP [25]. If, however, this increased cardiac response is insufficient to adequately perfuse the brain and heart (perhaps, for example, because of coexistent hypovolaemia), then splanchnic vasoconstriction would occur, allowing blood volume to be maintained in essential compartments at the expense of the gut. 

In the general population, common causes of OH include hypovolemia, polypharmacy, heart failure, arrhythmias, and advanced valvular heart disease [5]. In these conditions, both central and peripheral mechanisms are intact, in contrast with neurogenic OH, which is characterised by the pathological impairment of peripheral autonomic mechanisms. Neurogenic OH occurs in people with spinal cord injuries [26] and small fibre neuropathies, including diabetes [17]. While OH in PD is considered neurogenic in origin [5], it differs from other neurogenic OH because its pathophysiology is contributed to by the impairment of both peripheral and central mechanisms [20]. The baroreflex gain is low [27], indicating a dysfunctional central mechanism. PD pathology is present in the brain stem sites mediating the baroreflex [21,23] and also in the insular cortex [24] (see Ref. [20] for a review). While central control of sympathetic function is disturbed relatively early in PD [28,29], baroreflex failure alone does not usually cause OH [27], as peripheral mechanisms must also be present. Evidence for impaired peripheral mechanisms in OH of PD includes a low noradrenergic response to orthostatic challenge [30,31,32] and cardiac sympathetic denervation and dysfunction [33] (see [20,34] for a review). 

The consequence of this broader autonomic dysregulation in PD is that the combined effect of otherwise minor stressors, such as the vasodilating effect of levodopa, hypertensive agents, exercise, dehydration, and food [1,25,35], cannot be defended against. For example, consider a person who has breakfast in the morning when their BP is already low because of relative dehydration and levodopa-induced vasodilation. Impaired central mechanisms mean that post-breakfast splanchnic vasodilation cannot be inhibited and instead persists, further compounding low BP. Furthermore, cerebral perfusion may be further compromised because cerebrovascular autoregulation is also disturbed [4,36]. Thus, in PD, OH appears intermittently and often in response to a confluence of stressors. On the other hand, supine hypertension may occur [10] because the baroreflex and renal mechanisms are not centrally coordinated to respond to fluid from the lower extremities returning to central compartments overnight. There is also marked variability in systolic BP [37,38,39], which is frequently elevated in the morning. Capturing these features requires frequent BP measures with morning measurements or measurements when at least one stressor, for example, standing, is present.

In the routine clinical care of PD, OH is usually identified by a single lying and standing systolic BP in the clinic. Performed properly, this requires the person with PD (PwP) to lie resting for 5 min followed by BP measured supine, immediately when standing, and then 3 min later. This is a serious impost on time in a busy practice, and an anecdotal poll of colleagues in private practice suggests that compromises are made and corners are cut, even to the extent of measuring sitting rather than lying BP. Thus, an effective alternative would be welcomed in routine care. Measurement is often prompted by a history of symptoms consistent with OH. However, history is unreliable, with episodes of OH frequently asymptomatic or unrecognised by the PwP [4,37,40], as well as the presence of symptoms not correlating with the presence of OH [37]. Furthermore, as discussed above, OH can be intermittent and thus missed by a single measurement, which also cannot identify variability or morning supine hypertension. Twenty-four-hour blood pressure recordings are frequently used but they do not readily identify stressor-induced drops in BP and, as noted above, OH recognition is low in PD, so self-reporting diaries can fail. A novel pilot study of eight subjects undergoing continuous 5-day monitoring [37] provided results indicating that unrecognised events and systolic variability could be detected by prolonged recordings. Usual OH assessments are lab-based, expensive, and do not address the issues of OH in PD: particularly BP variability and supine hypertension.

This study was directed at the question of whether more frequent measures of lying and standing blood pressure performed at home by the PwP, including early morning measurement, might improve the detection of OH, supine hypertension, and systolic BP variability. PwPs were provided with a calibrated electric sphygmomanometer and were instructed in taking and recording lying and standing BP. They then took twice daily measurements on five consecutive days in their own home. The results were compared to lying and standing BP measured in the clinic. 

## 2. Materials and Methods

This study was approved and overseen by the St. Vincent’s Hospital (Melbourne) Human Research and Ethics Committee (approval number LRR 320.21). Subjects provided written consent according to the Declaration of Helsinki, and the study was conducted according to the International Conference on Harmonisation: Good Clinical Practice Guidelines (ICH-GCP).

### 2.1. Subjects and Recruitment

Participants were 44 PwPs with a history of idiopathic PD and 16 people without PD (controls: usually the spouse of the PwP). All participants were aged 60 years or more. PwPs were required to be 6 or more years from onset of symptoms or diagnosis to increase the likelihood that a significant proportion would have clinical OH at the time of enrollment (27% had postural drop in the clinic plus symptoms, as shown in Table 1) and that a similar proportion would not have OH, even on repeated measures. Cases with other potential causes of OH including a prescription for diuretics, diabetes (requiring insulin), small fibre neuropathy, heart failure, renal failure, or other reasons for fluid volume disturbance were excluded. Medications that could contribute to OH were recorded but, except for diuretics and insulin, were not a cause for exclusion. Antihypertensives were taken by 27% of the PwPs and 47% of the controls. Medications for urinary urgency were taken by 10% of the PwPs and 6% of the controls. Antidepressants were taken by 4% of the PwPs and 6% of the controls. Fludrocortisone was taken by 6% of the PwPs. PwPs were recruited by reviewing the clinic appointment diary to identify subjects who were due to attend the clinic and contacting them by phone to assess their eligibility and willingness to participate.

On the day of attendance at the clinic, written consent to participate was obtained. Participants were provided with instructions for recording lying and standing blood pressure at home (see below). Lying and standing blood pressure was also measured. Clinical scales were administered (see next section). Participants’ demographics, medications, and data from various clinical scales were recorded and are shown in Table 1.

### 2.2. Clinical Scales

Clinical scales performed included the Movement Disorder Society United Parkinson’s Disease Rating Scale (MDS-UPDRS), the Montreal Cognitive Assessment (MoCA), the Non-Motor Questionnaire [41] (NMS-Q), the Parkinson’s Disease Questionnaire (PDQ-39), and the Orthostatic Hypotension Questionnaire (OHQ).

### 2.3. Blood Pressure Recordings

Participants were provided with an Omron HEM 7121 electronic BP machine that was calibrated by the hospital’s biomedical engineering department. Instructions for recording BP were:Attach the cuff to the arm, lie horizontal for 5 min, and then record BP;While still wearing the cuff, stand immediately and record the BP;Measure twice a day (on awakening and before arising and at night before retiring);Only perform measurements in the presence of a carer and sit or lie on the bed immediately if a risk of falling is perceived;After each reading, record the systolic and diastolic pressures on the provided chart. To avoid bias, PwPs were not informed about the meaning of the BP parameters they recorded.

Both the partner and PwP were asked to attend the training session and nominate which of them would be responsible for the recordings. The carer took responsibility approximately 50% of the time, particularly when cognition of the PwP was affected. Subjects were shown how to perform the recordings and how to record the result on the chart provided. They were requested to perform BP recordings until competent. 

All control subjects and 84% of PwPs recorded BP on 5 consecutive days, with the remaining 16% making recordings on 4 of the 5 days. The difference between standing and lying systolic BP (ΔBP) was calculated: a positive number indicated standing BP > lying BP. While ΔBP described the difference between a single pair of measurements, there were 10 measurement pairs (ΔBP) made at home over 5 days. These were described by the median, 75th percentile (the 3rd highest of 10 ΔBP), and the maximum of the 10 measurements (notated as ΔBP_med_, ΔBP_75th_, and ΔBP_MAX_, respectively). A ΔBP equal to or greater than 20 mmHg was considered “high”. Systolic readings were defined as hypertensive if they were equal to or greater than 145 mmHg. 

### 2.4. Statistics

As most distributions did not pass the D’Agostino and Pearson normality test and populations were small, the null hypothesis for the two distributions was tested using the Mann–Whitney test or the Wilcoxon matched pairs signed-rank test when the data werepaired. Categorical comparisons were assessed using the chi-squared test (or Fisher’s exact test if the samples were small). Cohen’s kappa statistic was used to measure concordance between existing measures of orthostatic hypotension and those from 5 days of recording at home. Statistical significance was set at 0.05. 

## 3. Results

### 3.1. Characteristics of Morning and Evening Systolic BP Readings

The median lying systolic and diastolic BP of the PwPs and controls are shown in Table 1. However, aggregating the readings obscures detail revealed by examining all systolic BP readings (432 from the PwPs and 160 from the controls) (Figure 1A). As 85% of participants contributed ten readings and the minimum from any subject was eight readings, all participants provided similar amounts of data, and examining every recording (as in Figure 1A) was not biased by one individual’s data. The median systolic BP reading was hypertensive (≥145 mmHg) in 32% of the PwPs and 25% of the controls.

Morning and evening lying systolic BP from the same day were examined as a pair, leading to the following observations:
*Morning systolic lying pressures are higher than their evening pair* in both PwP pairs (67%) and control pairs (75%). The difference between morning and evening systolic pressures was significant for both the PwPs (median difference = 6 mmHg, *p* < 0.0001—Wilcoxon matched pairs signed-rank test) and the controls (median difference = 4 mmHg, *p* < 0.01—Wilcoxon matched pairs signed-rank test);*If the morning lying systolic BP was 20 mmHg higher than its evening pair, it was frequently hypertensive in PwPs (78%) but not controls (38%*). On the other hand, when the evening lying systolic reading was the highest of the pair, the morning systolic was below 145 mmHg (80% of the PwPs and 98% of the controls).

### 3.2. Orthostatic Effects on Systolic BP

Measurements of standing and lying BP in the morning and evening for 5 days at home were used to calculate the ΔBP_med_, ΔBP_75th_, and ΔBP_MAX_ as measurements for evidence of OH (Figure 1B). Two observations are apparent. First, the proportion of subjects with a high ΔBP (by any of the three measures) was greater in the PwPs than in the controls (15.6%, 33.3%, and 62.2% for the PwPs and 0%, 17.6%, and 35.3% for the controls, respectively). Because of the number of controls with an elevated ΔBP_MAX_, the effect of a higher threshold (for example, 25 mmHg being the 90th percentile of the controls) was also examined. The horizontal dotted line in Figure 1B shows this number and the number of cases whose ΔBP_MAX_ ≥ 25 is shown as the top number in the small boxes at the base of each graph.

Second, the variability in readings from the PwPs was greater than the variability of the controls (also apparent in Figure 1A). This variability was examined further by calculating the difference between the maximum and minimum morning and evening lying systolic BP (Syst BP_Var_), which was considerably greater in the PwPs (39.5 (IQR = 30.3)) than in the controls (*p* = 0.02: unpaired *t*-test with Welch’s correction). Syst BP_Var_ was plotted against ΔBP_MAX_ (Figure 1C), showing a modest relationship between the two measures (with Cohen’s κ = 0.51 (discussed further below). This suggests that Syst BP_Var_ might be a marker of autonomic dysregulation, so it was compared in subjects with and without hypertension (Figure 1D). The Syst BP_Var_ was significantly larger in the PwPs when the median morning lying systolic BP was hypertensive (*p* = 0.01, Mann–Whitney test); this was not apparent in the controls. The trend for a higher orthostatic drop in hypertensive PwPs was not significant (Figure 1D), although PwPs with a large ΔBP_MAX_ (≥20) had a higher systolic BP (147 (IQR = 39)) than those whose ΔBP_MAX_ was low (125 (IQR = 34), *p* <0.07 *t*-test).

The interrelatedness of ΔBP_MAX_, Syst BP_Var_, and systolic BP was further examined. Of the 28 (out of 44) PwPs with an elevated ΔBP_MAX_, 17 had a high Syst BPVar and 10 had hypertension. Hypertension or a high Syst BP_Var_ without a high BP_MAX_ was uncommon (9%). This suggests that these three measures are largely coincident. 

### 3.3. Measurement of ΔBP at Home Compared to the Clinic

Next, the single office-based measurement of ΔBP (ΔBP_CLIN_) was compared with ΔBP_75th_ and ΔBP_MAX_ (Figure 1E). The BP_CLIN_ was equal to or above 20 mmHg in 28.9% of the PwPs, which is a little less than ΔBP_75th_ (33.3%). ΔBP_75th_ was also a little better correlated with ΔBP_CLIN_ (Pearson’s ρ = 0.66 and Cohen’s κ = 0.48) than ΔBP_MAX_ (Pearson’s ρ = 0.58 and Cohen’s κ = 0.34). However, ΔBP_75th_ gave more “false negatives” (cases in the bottom right quadrant in Figure 1E where ΔBP_75th_ failed to detect the OH observed in the clinic) than ΔBP_MAX_, whose differences with ΔBP_CLIN_ were almost all “false positives” (cases in the upper left quadrant in Figure 1E where ΔBP_MAX_ detected OH which was not observed in the clinic). It seems more plausible that one of the ten measures (ΔBP_MAX_) would detect intermittent OH more accurately than either one of the seven measures (ΔBP_75th_) or a single random measure in the clinic. For this reason, ΔBP_MAX_ was compared with scores from various clinical scales.

### 3.4. Relationship between ΔBP_MAX_ and Scores from Clinical Scales

The relationships between responses to Q1.12 of the MDS-UPDRS (light headedness on standing) and ΔBP_MAX_ and ΔBP_75_ are shown in Table 2. There was a progressive (but not statistically significant) trend for ΔBP_MAX_ to increase with a higher score to Q1.12. It was significant that a little more than half of those who responded with a “0” to this question had an elevated ΔBP_MAX_ and 20% of those who responded with a “2” or “3” had ΔBP_MAX_ < 20. A higher Q1.12 score tended to be associated with a higher ΔBP_MAX_ (Figure 2A), even though ΔBP_MAX_ weakly predicted any answer of “1” or more to this question (Cohen’s κ = 0.23). As a higher total score on the NMS-Quest scales was also associated with a higher ΔBP_MAX_ (Figure 2A), the relationship between ΔBP_MAX_ and responses to NMS questions specific to autonomic dysfunction (5–9, 19, 20, and 28) was examined (Figure 2A). An MDS-UPDRS I (total) score of 10 or more was associated with a higher ΔBP_MAX_ (*p* = 0.026, Mann–Whitney; see Figure 2B). There was a statistically insignificant trend for a lower MoCA and higher UPDRS III score with high ΔBP_MAX_. No relationship was found between ΔBP_MAX_ and the PDQ39 or other MDS-UPDRS scores. 

## 4. Discussion

The aim of this study was to assess whether more frequent measures of BP would provide a better indication of the presence of OH in PD. It was not intended to be a study of the incidence of OH in PD. PwPs with six or more years of disease duration were recruited to ensure that the study cohort included PwPs with and without OH. In the context of the aim of the study, its outcome can be assessed from a narrow perspective of whether an orthostatic drop in systolic BP was present and from a broader perspective of whether the dysregulation of systolic blood pressure control was present (expressed as morning hypertension and systolic BP variability (Syst BP_Var_) and OH).

From a narrow perspective, multiple measurements were more likely to provide at least one measurement ≥ 20 mmHg (*n* = 28) than a single clinic measurement (*n* = 13). There was a significantly higher chance of having at least one elevated ΔBP in the PwPs than in the controls (*p* = 0.0026, Fisher’s exact). Much of the thinking about OH is influenced by findings in hypovolaemia and OH induced by antihypertensive agents, which also provided the origin of a “high ΔBP” being ≥20 mmHg. In that setting, OH is expected to be consistently present. In contrast, OH in PD is intermittent [37], possibly reflecting central dysregulation [1,36,37,38] and the coincidence of different stressors, such as enteric shunting following food, and pharmaceuticals such as levodopa, hot environments, exercise, or hypovolaemia. While it seems logical that more frequent measurement would detect OH, it is unclear whether the optimum number of measurements should be five in ten (ΔBP_MED_), three in ten (ΔBP_75th_), one in ten (ΔBP_MAX_), or even one in twenty. The findings of Polverino et al. [37] provide some indication that around 10 measurements may be sufficient, although all participants in that study had established OH. In this study, only 27% were recognised as having OH, yet 66% had at least one elevated ΔBP. Certainly, the association between ΔBP_MAX_ and worsening UPDRS I and NMS-Quest scores in this study suggests that OH detected by ΔBP_MAX_ is meaningful. Other than the study of Polverino et al. [37], we are not aware of a similar attempt to use conventional BP measurements of lying and standing BPs at home to assess the presence and severity of OH. The Polverino et al. study [37] used sophisticated telemetry, which may lend greater certainty to compliance, but the technology did not obviate the need for the PwP to interrupt their day for the length of time that is required to carry out lying and standing BPs. Also, they measured eight subjects with known OH and thus cannot provide an indication of the value of at-home measurements capturing milder forms of OH. 

From a broader perspective, 10 readings at home provided insights regarding morning hypertension and increased variation in systolic pressures. However, hypertension or a high Syst BP_Var_ without a high ΔBP_MAX_ was uncommon (9%) and so, while a high Syst BP_Var_ or hypertension in the presence of a high ΔBP_MAX_ gives support to the finding of cardiovascular dysregulation, either Syst BPVar or hypertension in isolation does not.

While this study gives overall support for the home measurement of ΔBP, it does produce outstanding questions. Ten home measurements were arbitrary, as was the choice to perform the measurements at the start and end of the day rather than after meals. However, previous reports suggest that PwPs may not be fully aware of the presence of OH and misinterpret other symptoms as OH [37]. And, while the response to Q12.1 broadly correlated with the median postural drop, it was possible for individual cases to be unaware of OH, whereas others over-reported it. Early morning was chosen to capture supine hypertension. Measuring at the start and end of the day was expected to provide good compliance, which was excellent in this study, whereas asking subjects to measure when symptomatic may lead to overlooked measurements or measurements in response to irrelevant symptoms. Most participants (or their carer) had little difficulty in correctly measuring lying and standing BP, although subjects with cognitive impairment would not have been able to participate without a supportive carer. In summary, 10 recordings at home were superior to a single office measure in identifying the presence of OH as well as dysregulation in terms of a high Syst BP_Var_. Nevertheless, the ideal number of measurements and their best timing throughout the day require further study.

We have used the conventional threshold of 20 mmHg or higher for an elevated ΔBP. Our definition was limited to a measurement immediately after standing, not 3 min later or even 10–30 min after standing. There was concern that a more complex measurement paradigm invited poorer compliance. Justification for the protocol used here was provided by the relationship between BP_MAX_ and the clinical scales, but in particular, those questions from the NMS Quest indicated autonomic dysfunction (Figure 2A). A systolic BP ≥145 mmHg was used to define hypertension. This was chosen as closer to the threshold that triggers concern in the real-world management of hypertension when OH is present, even though lower pressures may invoke intervention in otherwise healthy individuals. As morning hypertension was used as a proxy for supine hypertension, 145 mmHg may be too rigorous for some [42] but not for other [43] authorities.

Would early therapeutic intervention result in a more sensitive measure indicator of the presence of OH? While the relationship between OH and cognitive decline has led to recommendations of early interventions [44], it is not clear whether OH is a surrogate for supine OH (reviewed in [43]), although findings here suggest they usually co-exist. Others raise the possibility that OH and cognitive decline are para-phenomena [20]. 

In this study, people with insulin-dependent diabetes were excluded to avoid other causes of autonomic dysfunction. Participants using diuretics were excluded because diuretics might exacerbate hypovolaemia, but the use of antihypertensives was not grounds for exclusion. We acknowledge this is inconsistent, especially as the use of antihypertensives was higher in the controls than in the PwPs, suggesting that antihypertensives might have been explicitly avoided in PwPs. As diabetes and the use of diuretics and hypertensives are nearly ubiquitous in this age group, a future study could accept their presence because it reflects the complexity facing the management of Parkinsonian subjects with autonomic dysregulation and other conditions. Further studies on the concurrent management of OH and these conditions are required, especially for morning hypertension. 

### Limitations of This Study

Many of the limitations of this study have been addressed above. These and other limitations are summarised here:
Are 10 measures adequate or too few? Should measures at other times (e.g., postprandial) also be included?;What proportion of measures should be sufficient to identify OH: 50% (ΔBP_MED_), 33% (ΔBP_75th_), 10% (ΔBP_MAX_), 5%, or even less?;This study did not use the more stringent criteria for OH and systolic hypertension recommended by some authorities;This study excluded insulin-dependent diabetes and users of diuretics but not users of antihypertensive agents. However, because of the loss of ability to regulate vasodilation in the various vascular beds, it is these cases that introduce complexity to the management of OH in PD. Thus, future studies could examine the trade-off in treating hypertension in the presence of OH, especially when multiple measurements, such as those proposed here, are used;Although participants were trained to use the sphygmomanometers, we cannot exclude the possibility that some recordings were the result of poor technique or inaccurate recording. Poor technique might over-report hypotension and could also under-represent large postural drops. It is notable that very few systolic BP measures were less than 100 mmHg (Figure 1A);The sample size was large enough to show that at least one elevated ΔBP in 10 measurements is more likely to be found in PD than the controls. Larger samples would be required to address the dot points outlined above.


## 5. Conclusions

Twice daily recordings of lying and standing BP over 5 days increase the likelihood of finding an elevated postural drop, which is consistent with OH. Moreover, it can show whether there is increased variation in systolic pressures and morning hypertension, which is consistent with the dysregulated control of BP. PwPs were able to perform and record BP measurements without complications, were compliant, and did not find it intrusive. 

Further studies are required to establish whether 5 days of recording is sufficient and whether active intervention on finding dysregulated BP control results in better outcomes compared to waiting for symptomatic OH before intervening. 

## Figures and Tables

**Figure 1 jpm-13-01324-f001:**
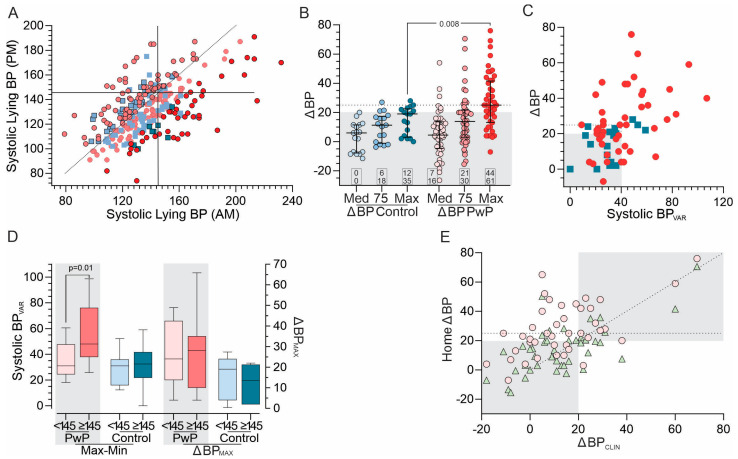
(**A**) A plot of the morning (AM, x-axis) and evening (PM, y-axis) lying systolic BP. Vertical and horizontal black lines indicate a reading of 145 mmHg and the dotted line indicates when morning and evening readings are the same. Pink circles show measurements from the PwP, with circles with a black border indicating cases where the evening reading was greater than the morning reading. Red circles indicate the PwP whose morning systolic measurement was ≥20 mmHg higher than the evening measurement. Grey squares show measurements from the controls, and those with a black border are cases whose evening reading was greater than the morning reading. Teal squares indicating cases where the evening reading was greater than the morning reading; (**B**) scatter plots (error bars: median and IQR) of the ΔBP (y-axis) sorted according to each participant’s ΔBP_med_, ΔBP_75th_, and ΔBP_MAX_ (each circle indicates an individual participant). The grey-shaded region represents a ΔBP of 20 mmHg, and the horizontal dotted line indicates ΔBP of 25 mmHg. At the base of each plot are two sets of numbers in a box: the lower number indicates the percentage of that category where the ΔBP ≥ 20 mmHg and the upper row indicates the percentage of that category where the ΔBP ≥ 25 mmHg. Only *p*-values < 0.05 (Mann–Whitney test) are shown; (**C**) a plot of ΔBP_MAX_ (y-axis) of the PwPs (red circles) and the controls (teal squares) against the difference between the Syst BP_Var_ (x-axis: maximum–minimum lying systolic BP). The grey shaded area represents the region where both the ΔBP_MAX_ < 20 and the Syst BP_Var_ < 40 mmHg (which is~ the 75th percentile of the controls; see (D)) are present; (**D**) box (median and interquartile range) and whiskers (10th and 90th percentile) representing the range of Syst BPVar (left y-axis: maximum–minimum systolic BP) and ΔBP_MAX_ (right y-axis) of the PwPs (pink and red boxes) and the controls (grey and teal). Only *p*-values < 0.05 (Mann–Whitney test) are shown; (**E**) a plot of ΔBP_75th_ (green triangles) and ΔBP_MAX_ (pink circles) on the y-axis against the ΔBP_CLIN_ (x-axis). Concordance between ΔBP_CLIN_ and the measurements at home are symbols in the lower left grey rectangle (no OH) and the upper right grey rectangle (OH). Symbols in the upper left quadrant show cases where the home measurement detected OH but the clinic measurement did not, whereas symbols in the lower right quadrant show cases where the clinic measurement found OH but the home measures did not.

**Figure 2 jpm-13-01324-f002:**
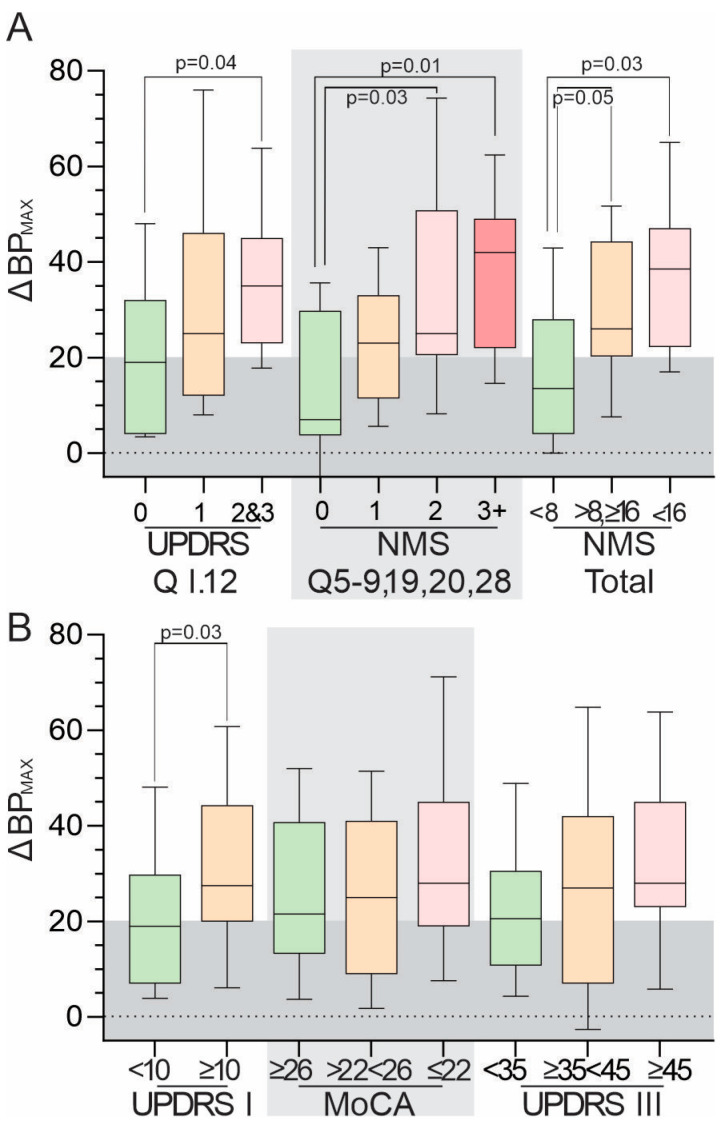
(**A**,**B**) are boxes (median and interquartile range) and whiskers (10th and 90th percentile), representing the ΔBP_MAX_ (y-axis) of various clinical scales in PwPs (abbreviations provided in the Methods). The grey shaded area indicates the region where the ΔBP_MAX_ < 20 mmHg is found. Only *p*-values < 0.05 are shown from ordinary one-way ANOVA (in cases of three sets of data) or the Mann–Whitney test (in the case of two sets of data).

**Table 1 jpm-13-01324-t001:** Participants’ demographics, BP, and data from clinical scales.

Parameter	Control	PwD
Age	69 (9)	72 (8)
MoCA	26 (3)	24 (5)
Systolic BP	128 (22)	131 (25)
Diastolic BP	74 (12)	77 (14)
Disease Duration		10 (6)
UPDRS I		11 (7)
UPDRS II		15 (12)
UPDRS III		40 (20)
UPDRS IV		6 (6)
UPDRS Total		60 (29)
MDS_H&Y		2 (1)
OHSA TOTAL		0 (3)
OHDAS TOTAL		0 (0)
PDQ 39		21 (44.5)
NMS TOTAL		12 (10)
Prior Diagnosis of OH	1/16 (6%)	12/44 (27%)

All values are the median with the interquartile range (IQR) in brackets. Abbreviations for the clinical scales are defined in Section 2.2.

**Table 2 jpm-13-01324-t002:** The relationships between responses to Q1.12 of the MDS-UPDRS and ΔBP_MAX_ and ΔBP_75_.

MDS-UPDRS Q1.12 Response	0	1	2	3
Number (%)	25 (57%)	9 (21%)	7 (16%)	3 (7%)
Median ΔBP_MAX_	21 (20.5)	29 (35)	35 (36)	42 (41)
Median ΔBP_75th_	9 (20)	19 (25)	26 (39)	19 (26)

0: Normal: No dizzy or foggy feelings. 1: Slight: Dizzy or foggy feelings occur. However, they do not cause me trouble doing things. 2: Mild: Dizzy or foggy feelings cause me to hold on to something, but I do not need to sit or lie back down. 3: Moderate: Dizzy or foggy feelings cause me to sit or lie down to avoid fainting or falling.

## Data Availability

The data are available upon reasonable request to the corresponding author.

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
