# Peer review of "At-Home Blood Pressure Measurements Provide Better Assessments of Orthostatic Hypotension in Parkinson’s Disease"

_jpm, 2023, doi:10.3390/jpm13091324_

Round 1
Reviewer 1 Report
The authors presented an interesting review about the question of whether more frequent measures of lying and standing blood pressure might improve the detection of orthostatic hypotension in patients with Parkinson's disease. I have some comments:
1) The limitations of the study, of which there are many, need to be discussed in separate section. First of all, the very small group size limits the validity of the orthostatic hypotension frequency assessment.
2) A more detailed description of the clinical characteristics of the patients in Table 1 is required.
3) It is unclear whether the presence of hypertension was assessed and how it was combined with orthostatic hypotension? Prior 24-hour BP monitoring data are very useful, or this is a significant limitation of the study.
4) Please list in Methods the full list of potential causes of orthostatic hypotension used as exclusion criteria.
5) How were BP data collected: did patients keep a diary or was an automated data collection system used?
6) Patient adherence to study design was 100%?
Author Response
The limitations of the study, of which there are many, need to be discussed in separate section.
A separate section in the discussion has been provided (Section 4.1 Limitations of the Study).
- First of all, the very small group size limits the validity of the orthostatic hypotension frequency assessment.
We are unclear which particular aspect of validity is being challenged because of the group size. First the likelihood of a PwP in this cohort having at least 1 measure where the drop in systolic BP ≥20 mmHg, is significantly greater than in a control (p=0.0026, Fishers exact). The question of whether 10 measurements is sufficient is addressed in the discussion, where we make the point that the number of measurements required would need a follow-up study. This discussion has been extended and clarified.
- A more detailed description of the clinical characteristics of the patients in Table 1 is required.
The results of all of the clinical scales are now added to Table 1.
- It is unclear whether the presence of hypertension was assessed and how it was combined with orthostatic hypotension? Prior 24-hour BP monitoring data are very useful, or this is a significant limitation of the study.
Hypertension was assessed based on the 10 measurements over 5 days and with one being an early morning measurement. Please note that Section 3.1, Section 3.2 paragraph 2, Figures 1A and 1D are entirely devoted to systolic BP, the proportion that were hypertensive and the relation of hypertension to start or end of day. Variability of systolic measures is also assessed (see second para of Section 3.2).
It is also reviewed in the Results, along with the relationship between OH, morning BP elevation and systolic BP variation.
- Please list in Methods the full list of potential causes of orthostatic hypotension used as exclusion criteria.
These are listed in Section 2.1.
- How were BP data collected: did patients keep a diary or was an automated data collection system used?
PwP were provided with a standard form and at training were shown how to transfer the relevant numbers from the machine to the form. This has now been added to Section 2.3.
- Patient adherence to study design was 100%?
Line 166 (Section 2.3 in Methods) stated that “All control subjects and 84% of PwP recorded BP on 5 days with the remaining 16% making recordings on 4 of the 5 days.”.
Reviewer 2 Report
The article presents a comprehensive study investigating the potential benefits of frequent home measurements in detecting orthostatic hypotension (OH) in Parkinson's Disease (PD) patients. The study is well-structured and provides insightful findings that contribute to the understanding of OH in PD. The author effectively highlights the significance of OH in PD and its potential impact on cognitive decline. The methodology is well-explained, allowing for replication and understanding of the study design. The statistical analyses are appropriately applied, enhancing the reliability of the results.
Major Comments
1. The introduction could benefit from an expanded discussion on the shortcomings of current clinical practices and how the proposed method may enhance them. Including a thorough analysis of the study's significance would add depth to this section.
2. Consider restructuring the introduction to avoid hinting at the study's conclusions ("The findings suggest that multiple home measurements... will result in better detection..."). While this may be common practice in some fields and please consider clearly stating the research problem in the introduction.
3. The method of recording blood pressure needs further clarification. Include information on standardization, communication to subjects, and follow-up procedures to ensure process uniformity and compliance.
4. The decision to include participants with PD who had six or more years of experience should be further explained.
5. The manuscript should explicitly state that the study focuses on testing the utility of home BP measurement for detecting OH and is not an analysis of OH incidence in PD or its onset.
6. You may comment on the broader literature in this area while mentioning the findings of Polverino et al. [6]. Compare the findings of your study to those of other similar studies, emphasizing similarities and differences in techniques, findings, and interpretations.
7. While discussing the limitations of the inclusion criteria and measurement frequency, you may go into further detail on potential sources of bias or confounding factors that could impair the study's validity or generalizability. This demonstrates that you are aware of the study's limits and potential difficulties.
Minor Comments
1. Terms like "broader autonomic dysregulation" may be considered ambiguous without specific definitions. Please provide clear explanations for such terminology to ensure reader comprehension.
2. The abstract was full of small mistakes causing confusion about the purpose of this study from the beginning, examples include:
a. Line 16 “Participants were instructed in measure lying and standing BP when arising”, it is supposed to be “to measure”.
b. Line 20 had a starting bracket in the middle of the sentence without an ending.
c. The last 4 lines in the abstract will not be clearly understood for a first-time reader of the article.
d. The keyword “orthostatic hypertension” is supposed to be “orthostatic hypotension”.
Considering the importance of the abstract in providing the initial idea of the following article, small details should be attended for in order to give the reader the right overview.
3. The introduction included multiple english mistakes, including:
a. Line 32, it’s “lightheadedness” instead of “lighted headedness”.
b. Line 54, it’s clearer to use a dash in “orthostatic-induced fall”.
c. Line 43, “Apart from falls due to an episodic loss of consciousness, the identification of OH is important” should be shortened as this information was previously mentioned. The sentence can begin with “The identification of…”.
4. The instructions in lines 81, 82, 83, 84 should start with capital letters “Attach, While, Measure, Only”.
a. The sentences should also end with a dot as they are considered separate sentences.
b. The brackets in line 83 should include a comma to explain when exactly are “twice per day”.
5. Lines 113 and 114 is a very confusing sentence, “no individual biased the data when individual measurements were presented”.
6. Lines 235 and 236 should’ve been emphasized and explored more throughout the article, as they are considered high-yield information related to the topic of the article.
7. The paragraph starting from line 248 did not use formal professional English language, as well as included multiple grammatical mistakes:
a. Line 255, “excellent ion this study”.
b. Line 259, “these recordings did appear to identify”. It is more professional to just say “These recordings identify”.
c. Line 261 should be “outcomes”.
d. Line 266 “OH begins earlier than found using” is confusing.
8. Line 284, “likelihood” should be corrected. Line 286 should be “increased variation”. Line 287 should be “difficulties” instead.
The text includes multiple english mistakes, it needs to be fully revised.
Author Response
Major Comments
- The introduction could benefit from an expanded discussion on the shortcomings of current clinical practices and how the proposed method may enhance them. Including a thorough analysis of the study's significance would add depth to this section.
The Introduction has been completely rewritten and Addresses Reviewer’s points 2, 3, 5 and 6.
- Consider restructuring the introduction to avoid hinting at the study's conclusions ("The findings suggest that multiple home measurements... will result in better detection..."). While this may be common practice in some fields and please consider clearly stating the research problem in the introduction.
This has been removed.
- The method of recording blood pressure needs further clarification. Include information on standardization, communication to subjects, and follow-up procedures to ensure process uniformity and compliance.
This method of recording blood pressure has been expanded in Section 2.3.
- The decision to include participants with PD who had six or more years of experience should be further explained.
In Discussion and also in Methods.
- The manuscript should explicitly state that the study focuses on testing the utility of home BP measurement for detecting OH and is not an analysis of OH incidence in PD or its onset.
This was explicitly stated in the Discussion and has now been added to the introduction
- You may comment on the broader literature in this area while mentioning the findings of Polverino et al. [6]. Compare the findings of your study to those of other similar studies, emphasizing similarities and differences in techniques, findings, and interpretations.
Polverino et.al was mentioned in both Introduction and Discussion – there is a more extensive discussion in the Discussion section contrasting and comparing their findings with this study.
- While discussing the limitations of the inclusion criteria and measurement frequency, you may go into further detail on potential sources of bias or confounding factors that could impair the study's validity or generalizability. This demonstrates that you are aware of the study's limits and potential difficulties.
A new section (4.1. Limitations of this study.) has been added to the Discussion which addresses these concerns.
Minor Comments
- Terms like "broader autonomic dysregulation" may be considered ambiguous without specific definitions. Please provide clear explanations for such terminology to ensure reader comprehension.
The term has been removed.
- The abstract was full of small mistakes causing confusion about the purpose of this study from the beginning, examples include:
- Line 16 “Participants were instructed in measure lying and standing BP when arising”, it is supposed to be “to measure”.
- Line 20 had a starting bracket in the middle of the sentence without an ending.
- The last 4 lines in the abstract will not be clearly understood for a first-time reader of the article.
- The keyword “orthostatic hypertension” is supposed to be “orthostatic hypotension”.
Considering the importance of the abstract in providing the initial idea of the following article, small details should be attended for in order to give the reader the right overview.
These have been addressed.
- The introduction included multiple english mistakes, including:
- Line 32, it’s “lightheadedness” instead of “lighted headedness”.
- Line 54, it’s clearer to use a dash in “orthostatic-induced fall”.
- Line 43, “Apart from falls due to an episodic loss of consciousness, the identification of OH is important” should be shortened as this information was previously mentioned. The sentence can begin with “The identification of…”.
The Introduction has been completely rewritten can attention has been paid to spelling/English errors.
- The instructions in lines 81, 82, 83, 84 should start with capital letters “Attach, While, Measure, Only”.
- The sentences should also end with a dot as they are considered separate sentences.
- The brackets in line 83 should include a comma to explain when exactly are “twice per day”.
Changed as requested.
- Lines 113 and 114 is a very confusing sentence, “no individual biased the data when individual measurements were presented”.
This sentence now reads: “As all participants provided similar number of PB recordings, examining every recording (as in Figure 1a) is not biased by one individual’s data”.
- Lines 235 and 236 should’ve been emphasized and explored more throughout the article, as they are considered high-yield information related to the topic of the article.
This has been addressed in detail in the Introduction.
- The paragraph starting from line 248 did not use formal professional English language, as well as included multiple grammatical mistakes:
- Line 255, “excellent ion this study”.
- Line 259, “these recordings did appear to identify”. It is more professional to just say “These recordings identify”.
- Line 261 should be “outcomes”.
- Line 266 “OH begins earlier than found using” is confusing.
These have been corrected as suggested.
- Line 284, “likelihood” should be corrected. Line 286 should be “increased variation”. Line 287 should be “difficulties” instead.
Corrected as suggested.
Round 2
Reviewer 2 Report
No further comments.